# Does Complement-Mediated Hemostatic Disturbance Occur in Traumatic Brain Injury? A Literature Review and Observational Study Protocol

**DOI:** 10.3390/ijms21051596

**Published:** 2020-02-26

**Authors:** Alexander Fletcher-Sandersjöö, Marc Maegele, Bo-Michael Bellander

**Affiliations:** 1Department of Neurosurgery, Karolinska University Hospital, 171 76 Stockholm, Sweden; bo-michael.bellander@sll.se; 2Department of Clinical Neuroscience, Karolinska Institutet, 171 77 Stockholm, Sweden; 3Department for Trauma and Orthopedic Surgery, Cologne-Merheim Medical Center (CMMC), University Witten/Herdecke, 58455 Cologne, Germany; marc.maegele@t-online.de; 4Institute for Research in Operative Medicine, University Witten/Herdecke, 58455 Cologne, Germany

**Keywords:** traumatic brain injury, hemostasis, coagulation, coagulopathy, thrombosis, complement, inflammation

## Abstract

Despite improvements in medical triage and tertiary care, traumatic brain injury (TBI) remains associated with significant morbidity and mortality. Almost two-thirds of patients with severe TBI develop some form of hemostatic disturbance, which contributes to poor outcome. In addition, the complement system, which is abundant in the healthy brain, undergoes significant intra- and extracranial amplification following TBI. Previously considered to be structurally similar but separate systems, evidence of an interaction between the complement and coagulation systems in non-TBI cohorts has accumulated, with the activation of one system amplifying the activation of the other, independent of their established pathways. However, it is not known whether this interaction exists in TBI. In this review we summarize the available literature on complement activation following TBI, and the crosstalk between the complement and coagulation systems. We demonstrate how the complement system interacts with the coagulation cascade by activating the intrinsic coagulation pathway and by bypassing the initial cascade and directly producing thrombin as well. This crosstalk also effects platelets, where evidence points to a relationship with the complement system on multiple levels, with complement anaphylatoxins being able to induce disproportionate platelet activation and adhesion. The complement system also stimulates thrombosis by inhibiting fibrinolysis and stimulating endothelial cells to release prothrombotic microparticles. These interactions see clinical relevance in several disorders where a deficiency in complement regulation seems to result in a prothrombotic clinical presentation. Finally, based on these observations, we present the outline of an observational cohort study that is currently under preparation and aimed at assessing how complement influences coagulation in patients with isolated TBI.

## 1. Introduction

Despite improvements in medical triage and tertiary care, traumatic brain injury (TBI) remains associated with significant morbidity and mortality [1]. Up to two-thirds of patients with severe TBI develop complicating hemostatic disturbance, which further contributes to poor outcome and death [2]. TBI also results in increased complement activation and activity [3,4,5]. In non-TBI cohorts, there is growing evidence to support crosstalk between the complement and coagulation system, resulting in the amplification of their otherwise targeted responses [6,7,8,9,10,11,12,13,14,15,16,17,18,19,20,21,22,23,24,25,26,27,28,29,30,31,32,33,34,35,36,37,38,39,40,41,42,43]. However, it is not known whether this process also occurs in TBI. In this article, we summarize the available literature on complement activation following TBI and the crosstalk between the complement and coagulation systems. We also present the outline of an observational cohort study that is currently under preparation and aimed at assessing if complement activity influences coagulation in patients with isolated TBI.

## 2. Complement and Coagulation

### 2.1. Hemostatic Disturbance in TBI

The initial injury in TBI is often followed by secondary insults. One of these is hemostatic disturbance, defined as a defect in hemostasis that leads to an increased susceptibility to bleeding or thrombosis, which is present in up to two-thirds of patients with severe TBI and is independently associated with an increased risk of morbidity and mortality. In fact, many consider disturbed hemostasis to be the leading cause of preventable death following TBI. While it is still unclear exactly how TBI affects the coagulation system, the principle drivers in this context seem to be platelet dysfunction, endothelial activation, disturbed fibrinolysis, endogenous anticoagulation, and inflammation [2,44,45].

Currently, there is controversy within the literature regarding the exact nature of hemostatic perturbations after TBI, and evidence exists for the presence of both a hypercoagulable and hypocoagulable state [2]. For example, while the initial head injury often leads to increased bleeding tendency and the exacerbation of hemorrhagic lesions [2,46,47,48], TBI is also independently associated with an increased risk for venous thromboembolism [49,50,51,52] and ischemic stroke [53,54,55,56,57]. Autopsy studies have also revealed that micro-thrombosis is present in the majority patients who have died from head trauma [58]. Most likely, there is a progression from early clot formation to unregulated thrombosis culminating in a final consumptive coagulopathy that may later turn into increased risk for thrombosis again, but currently an overlap and lack of distinction exists between the phases.

Thus, the coagulation system is activated following TBI, which can result in a conflicting state of both coagulopathy and thrombosis. Regardless of the confusion, it is clear that the addition of hemostatic disturbance, however defined, contributes to poor outcome for these patients.

### 2.2. The Complement System

The complement system is part of the innate immune response [59,60] and consists of serine proteases that are encoded from the same ancestral genes as coagulation proteins [61]. It functions as a form of intravascular surveillance that, when activated, amplifies and forms the complement cascade. Much like the coagulation system, complement activation involves several highly regulated steps with the interaction of both plasma and membrane-bound proteins. The established function of the complement system is to eliminate foreign pathogens and substances, as well as to remove debris and dead cells. This is accomplished by tagging foreign surfaces with opsonins, generating pro-inflammatory mediators, and activating the membrane attack complex (MAC, also known as the terminal complement complex C5b-9) [62] (Figure 1).

The complement system is mainly activated by three pathways: the lectin pathway, the classical pathway, and the alternative pathway. Each is triggered by different agents, which converge in the formation of C3 convertase. C3 convertase then cleaves C3 into C3b, a potent opsonin, and C3a, an anaphylatoxin [62]. Further propagation of C3b also results in the generation of C5 convertase, which cleaves C5 to C5a, an anaphylatoxin capable of recruiting inflammatory mediator cells, and C5b. C5b then forms a complex with other complement proteins that make up the MAC. The deposition of the MAC on cell membranes leads to calcium influx and cell lysis but can also activate intracellular signaling at lower doses [62] (Figure 1).

To prevent over-amplification, the complement system is also regulated by complement control proteins. These include the C1-inhibitor (CI-INH), which can deactivate C1r and C1s of the classical pathway, as well as the C4-binding protein (C4BP), the decay-accelerating factor (DAF, also known as CD55), and factor H, which are able to break up C3 convertase and act as cofactors for the cleavage of C3b and C4b to their inactive degradation products [62,63].

### 2.3. The Complement System in the Healthy and Injured Brain

It is well known that components of the complement cascade are present within the brain [4,59]. Both neurons and glial cells are capable of synthesizing complement proteins [64], and complement receptor expression is widespread within the central nervous system (CNS) [65]. In the healthy brain, the complement system is involved in the clearance of cellular debris and apoptosis, and protecting from infection and inflammation [66].

Traumatic brain injury (TBI) has been shown to activate the complement system in the CNS [3,67,68], further intensified by an influx of circulating components following a breakdown of the blood–brain barrier (BBB) [59]. An early study from our group showed that elevated levels of complement can be found in the penumbra of human contused brain tissues [3]. Later studies have revealed that patients exposed to traumatic brain injury have increased concentrations of complement factors in both sera [69] and cerebrospinal fluid [5,70]. Intrathecal levels of complement factors following TBI also seem to correlate to the degree of BBB dysfunction [5], which, in conjunction with our finding that levels of complement factors following severe isolated TBI seem to be higher in arterial than jugular venous blood [69], could indicate that these changes are most likely due to altered blood–brain barrier integrity and not due to intrathecal synthesis. In a porcine model of TBI, the systemic activation of coagulation and complement is also evident within 3 minutes of injury [68]. In other preclinical studies, C3-deficient mice, as well as rats treated with complement-inhibitors, show reduced neutrophil infiltration, microglial activation, and edema formation at the site of injury [71,72]. Inhibiting the function of complement anaphylatoxins further reduces secondary damage in an experimental model of TBI [73]. In addition, the administration of an MAC complex formation inhibitor reduced neuronal damage and microglial activation in mice, suggesting that site-targeted complement inhibition could promote recovery [74].

Thus, complement is present both in and outside the CNS and is activated following isolated TBI. The question of whether elevated intracranial levels of complement factors are derived from intrathecal or systemic complement activation, with consecutive leakage across a dysfunctional BBB, remains to be evaluated.

## 3. Crosstalk between Complement and Coagulation

In this section, we present the available evidence supporting crosstalk between the complement system and the coagulation cascade, platelets, the fibrinolytic system, and endothelial cells.

### 3.1. Complement and the Coagulation Cascade

Studies have shown that TBI in itself triggers the coagulation pathway, resulting in initial thrombus formation and the later consumption of clotting factors [2]. One of the principal driving forces behind the coagulation cascade is the exposure to tissue factor, which initiates the intrinsic coagulation pathway. C5a, an anaphylatoxin of the complement cascade, has been shown to increase tissue factor activity in both circulating forms [6] and on endothelial cells [7]. This finding is supported by two ex-vivo studies demonstrating that the inhibition of C3 or C5 leads to a reduced expression of tissue factor, and in all probability also leads to a decrease in coagulation cascade activity [18,29]. The same reaction can be seen on mast cells, a type of granulocyte that is a part of the immune system, where the complement system is able to modify cell-surface phospholipid membranes to induce the expression of the tissue factor, thereby promoting a prothrombotic phenotype [38]. Thus, C5a can activate the intrinsic pathway of the coagulation cascade by increasing tissue factor activity (Table 1, Figure 2).

Mannan-binding lectin serine proteases (MASPs) are a group of characteristic serine proteases that initiate the lectin complement pathway. About a decade ago, Krarup and colleagues demonstrated that both MASP-1 and MASP-2 can directly cleave prothrombin to form activated thrombin, which can then form cross-linked fibrin [39]. This finding proves that the complement system is able to generate the end-product of the coagulation cascade independently of coagulation factors. More recent in-vitro studies have further shown that MASP-1, which shares many structural features with thrombin [40], can autonomously activate fibrinogen and factor XIII (fibrin stabilizing factor) [40,41]. Highlighting the possible clinical significance of these results, MASP-1 knockout mice show significantly longer tail-bleeding times compared to controls [42]. Another recent study, in which whole blood was perfused in a microchamber coated with endothelial cells, found that the addition of MASP-1 gave a significantly greater deposition of fibrin on endothelial cells [43]. Thus, MASPs of the lectin complement pathway seem able to produce thrombin independently of other coagulation factors, thereby providing a direct link between complement and fibrin formation (Table 1, Figure 2).

Complement system inhibitors also seem to be able to inhibit the coagulation cascade [8]. C1 esterase inhibitor (C1-INH), whose main function is to prevent the spontaneous activation of the complement system, has been shown to inhibit factor XII of the intrinsic coagulation pathway [9], as well as thrombin directly [10]. The C4b-binding protein (C4BP), which normally acts as a regulator of the classical and lectin complement pathway, can also inhibits protein S, a co-factor for the activated protein-C pathway of coagulation inhibition. Thus, the pro-coagulant activities of complement is further supported by the complement-inhibitor-mediated inhibition of the coagulation cascade [11] (Table 1, Figure 2).

In addition to complement-mediated activation and inhibition of the coagulation cascade, evidence in turn shows that the coagulation cascade can independently cleave complement proteins [12]. Most notably, thrombin, as well as factor Xa and factor XIa, can activate C3 and C5 independently of C3 convertase [12,13,14,15]. However, the kinetics of this cleavage may be poor [16], and it remains unclear if the thrombin-mediated cleavage of C3 and C5 is physiologically meaningful. Further excluding a central role for thrombin in complement synthesis, a murine model of venous thrombosis found the levels of the thrombin–antithrombin complex (TAT), which is a marker for thrombin generation, to be poorly correlated to C3a and C5a. In addition to the thrombin-mediated activation of C3 and C5, factor XIIa is able to activate C1r and thereby initiate the classical pathway of complement activation [12]. The thrombin-activated fibrinolytic inhibitor (TAFI) can also activate C3a and C5a in a negative feedback loop [17]. Thus, factors from the coagulation cascade can stimulate complement activity, although it is unclear if this has any clinical relevance.

In summary, the complement system is capable of activating the coagulation cascade on several levels. The primary facilitators behind this appear to be the C5a-mediated activation of the intrinsic coagulation pathway and the MASP-mediated production of thrombin.

### 3.2. Complement and Platelets

Evidence shows that TBI is followed by platelet dysfunction [75,76] and a decreased platelet count [77], both of which appear to increase the risk of bleeding [76,78,79].

Early studies showed that C3- and C5-deficient mice had prolonged tail-bleeding times and reduced platelet function [19], lending support to the notion that complement activates platelets. Studies on the molecular interaction between the complement and coagulation systems have shown that the MAC, one of the end-products of the complement cascade, is able to activate platelets and enhance platelet aggregation [20,21,22]. More recent studies have also shown that C3 deficiency in mice causes a prolonged bleeding time, reduced thrombus incidence and size, reduced fibrin and platelet deposition, and reduced platelet activation [23]. Other studies have further demonstrated that platelets have receptors for C3a that in turn can mediate platelet activation [24]. Thus, both C3 and the MAC of the complement system appear capable of activating platelets (Table 1, Figure 2).

The assembly of the MAC on human platelets has also shown to result in a dose-dependent increase in the binding of coagulation factors Va and Xa, which in turn results in an increase of platelet prothrombinase activity. Further evidence suggests that this MAC assembly on platelet surfaces initiates the release of factor V from alpha-granules [25], which are then able to generate procoagulant activity [26]. Removal of external Ca^2+^ seems to inhibit this MAC-initiated release of the platelet alpha-granule storage pool, suggesting that the effects that lead to increased platelet prothrombinase activity are mediated in part by influx of Ca^2+^ across the MAC pore [26]. Thus, the MAC also seems to be able to trigger platelet-dependent procoagulant activity through the release of microparticles (Table 1, Figure 2).

In an interesting study involving 40 trauma patients and 30 healthy controls, platelets from controls were incubated with sera from trauma patients, and platelet function as well as complement deposition on platelets was measured. The study found that complement activation increased platelet aggregation, and that platelets from trauma patients had significantly higher amounts of C3a and C4d on their surfaces compared to controls. However, despite this increased complement content, trauma sera rendered platelets hypoactive, with the depletion of complement from the sera further blocking the activation of the hypoactive platelets [80]. This finding is surprising, as one would expect the increased complement deposition in trauma platelets to result in increased platelet activity and aggregation. Presumably, trauma also initiates other processes that render platelets hypoactive despite the increased complement activity. The same pattern can also be seen in patients with TBI, where platelet function decreases in the first hours to days following injury [75]. Strengthening the hypothesis that complement activation leads to increased platelet activity, other studies have demonstrated that there is a correlation between MAC levels and plasma concentrations of prothrombin fragments 1 and 2, which are produced upon thrombin generation, in trauma patients with low C3a levels [27]. Treatment with a C3 inhibitor in primates with traumatic shock also results in reduced coagulation [28].

As was the case with the coagulation cascade, platelets also seem able to regulate complement activity. Del Conde et al. demonstrated that C3b binds to activated platelets by using P-selectin (CD62P) as a receptor, and proposed this as a mechanism by which alternative pathway activation occurs on the surface of activated platelets [30]. Chondroitin sulphate A (CS-A), which is released from platelet alpha granules during platelet activation, can also induce inflammatory signals mediated by C5a [31]. Complement regulators such as factor H, C1-inhibitor, and C4BP are also stored in the alpha granules that can be secreted from platelets, showing that platelets can activate the alternative and classical pathway of complement upon activation via secretory signaling [32,33]. Platelets also possess binding sites for C1q that, together with C1r and C1s, form the C1 complex that initiates the classical complement pathway [34]. Activated platelets and fibrin have also been shown to activate MASPs during blood clotting both in vitro and in vivo [35].

In summary, evidence points to a relationship between complement and platelets on multiple levels that can result in disproportionate platelet activation, as well the release of platelet-derived microparticles that further stimulates this reaction.

### 3.3. Complement and Fibrinolysis

The fibrinolytic system is made up of the inactive plasminogen, which can be converted to the active plasmin that then breaks down fibrin. Plasminogen is activated by plasminogen activators, which, in turn, can be inhibited by plasminogen activator inhibitors. In addition to decreased platelet function and the activation of the coagulation cascade, TBI also seems to induce immediate fibrinolysis [81,82], which appears to be a driving force behind trauma-induced disseminated intravascular coagulation (DIC), which is often followed by fibrinolysis shutdown, commonly seen hours to days after initial injury [83,84], and which could be a mechanism behind the increased risk of thromboembolism seen in TBI patients [85].

There is evidence pointing to interactions between the complement and fibrinolytic system. In a landmark study by Brown et al, C1-INH in its native state was found to inhibit plasmin [36], which could presumably lead to decreased fibrinolysis and thus increased thrombus formation. In the same manner, complement factors can induce the expression of plasminogen activator inhibitor-1 (PAI-1) by mast cells, further inhibiting fibrinolysis and thereby promoting a prothrombotic phenotype [38]. In vitro-studies also showed that MASP-1 can autonomously activate the thrombin-activated fibrinolytic inhibitor (TAFI, an inhibitor of fibrinolysis) [40,41] (Table 1, Figure 2).

Similar to the coagulation cascade and platelets, factors of fibrinolysis seem able to modify complement activity as well. The most important finding is probably that plasmin can activate C3 and C5 independently of C3 convertase [12,13,14,15]. In fact, plasmin-activated C5 can yield functional MACs, with the plasminogen activator increasing C5a levels. This means that plasmin bridges thrombosis and the immune response by liberating C5a and inducing MAC assembly [37]. Plasmin, therefore, in addition to thrombin, seems to be a key C5a generating enzyme.

In summary, fibrinolysis can be inhibited by both complement factors and complement inhibitors. This is done by limiting the formation of plasminogen to plasmin and stimulating endogenous fibrinolysis inhibitors, respectively.

### 3.4. Complement and Endothelial Activation

Endothelial activation also plays a part in TBI-associated hemostatic disturbance [2]. Evidence now shows that the complement system may also play a role in this process. For example, the MAC is able to induce endothelial cells to secrete von Willebrand factor, which plays a major role in blood coagulation [86]. While the precise mechanism behind this remains unclear, the assembly of the MAC on endothelial cell membranes appears to result in an influx of Ca^2+^ across the plasma membrane, which in turn leads to an increase in endothelial cytosolic Ca^2+^ and the secretion of platelet adhesive von Willebrand factor [86]. This MAC-induced secretion also seems to be accompanied by an increase in prothrombinase activity. The capacity of the MAC to induce the exposure of the prothrombinase enzyme complex may contribute to fibrin deposition, which is associated with immune endothelial injury [87]. C5a has also been found to induce a dose-dependent expression of endothelial P-selectin similar to that of thrombin [88]. This P-selectin is, in turn, important for the recruitment and aggregation of platelets to areas of vascular injury through platelet–fibrin and platelet–platelet binding (Table 1, Figure 2).

In summary, the complement system can also induce thrombosis by inducing endothelial cells to release von Willebrand factor and express P-selectin.

### 3.5. Clinical Examples of Complement-Mediated Hemostatic Disturbance

Clinically relevant interplay between the complement and coagulation system is highlighted by several disorders where a deficiency in complement regulation appears to result in a prothrombotic clinical presentation.

Paroxysmal nocturnal hemoglobinuria (PNH) is a rare hematological disorder associated with an acquired deficiency in the synthesis of glycophosphatidylinositol (GPI) that renders erythrocytes susceptible to complement-mediated destruction [89]. PNH is also associated with an increased risk of thrombosis linked to complement-mediated platelet activation [90]. Atypical hemolytic uremic syndrome (aHUS) is a rare, life-threatening disease caused by chronic, uncontrolled, and excessive activation of the complement system due to production of anti-factor H autoantibodies or genetic mutations in complement regulatory proteins. This complement overactivation results in platelet activation, leading to thrombocytopenia and thrombotic microangiopathy [91]. Moreover, in acute stroke, complement deposition on platelets has been linked to stroke severity and infarct volume [92]. In patients with systemic lupus erythematosus (SLE), increased complement deposition on platelets occurs in those with a history of venous thromboembolism as compared to controls [93]. These disorders collectively demonstrate that increased complement activity can trigger thrombotic events.

Eculizumab, a terminal complement inhibitor, is the first complement-specific drug to be approved recently by the US Food and Drug Administration and has been a therapeutic revolution for patients with aHUS and PNH. The agent, an anti-C5 antibody that blocks formation of the MAC and C5a generation, has been shown to reduce thromboembolic events in patients with PNH [94] and aHUS [95]. According to case reports, Eculizumab has also shown clinical benefit in treating thrombotic microangiopathy secondary to sepsis-induced DIC [96], as well as provide potential benefits in antiphospholipid syndrome [42]. Several other complement-inhibiting drugs for the treatment of coagulation disorders have entered late-stage pre-clinical and clinical trials [97,98,99,100,101].

## 4. Discussion

The aim of this study was to review the literature on complement activation following TBI, as well as the crosstalk between the complement and coagulation systems. We found that pre-clinical and clinical evidence suggests a range of interactions between the two systems, with the activation of one amplifying the activation of the other, independent of their respective established pathways.

With regards to the coagulation cascade, complement factors were shown to both increase tissue factor activity [6,7,18,29,38], thereby activating the extrinsic coagulation pathway, and form activated thrombin from prothrombin [39,40,41]. Complement system inhibitors were also able to inhibit the intrinsic coagulation pathway, thrombin activity, and protein S [9,10,11]. In addition to the coagulation cascade, complement factors were found to increase platelet activity and aggregation [20,21,22,23,24], prothrombinase activity, and the release of platelet-derived procoagulant granules [25,26], as well as stimulate endothelial cells to release prothrombotic von Willebrand factor and express P-selectin [86,87,88]. This evidence collectively suggests that increased complement activity leads to increased coagulation cascade activity and platelet aggregation, i.e., a prothrombotic state. This is further exemplified by a number of disorders (“complementopathies” [102]), such as PNH and aHUS, where inappropriate activation of the complement pathways results in thrombotic complications that can be successfully reduced with complement-inhibiting treatment.

Complement was also found to regulate fibrinolysis, with complement cascade inhibitors demonstrating the ability to inhibit plasmin [36], and complement factors able to activate the fibrinolysis inhibitors PAI-1 and the TAFI [38,40,41]. Somewhat contradictively, this shows that fibrinolysis can be inhibited by both complement factors and complement inhibitors.

With regards to TBI, there are no studies that have assessed if complement plays a role in the hemostatic disturbance that is seen following injury. Based on the data provided in this review, one might instinctively suggest that any such interaction would result in increased thrombosis, and could thus explain the prothrombotic sate that is seen days to weeks following TBI [53,54,55,56,57]. However, it should be noted that the initial bleeding tendency that is seen following TBI is also believed to be due to unregulated thrombosis, culminating in a consumptive coagulopathy that leads to the exacerbation of hemorrhagic lesions [2]. While this process is most often attributed to a trauma-induced release of brain-derived tissue factor that overactivates the extrinsic coagulation pathway [2], some part of this could be explained by the complement overactivation that is seen within minutes of TBI [68]. However, this remains to be answered. Another driving force behind TBI-induced bleeding is also hyperfibrinolysis [47,103], exemplified in the recently published CRASH-3 trial on the effect of tranexamic acid on death, disability, vascular occlusive events, and in patients with acute TBI [104], where one would expect clinically significant complement overactivation to lead to hypofibrinolysis instead. However, hypofibrinolysis following TBI is usually not seen until a few hours after initial injury [82,84], as opposed to hyperfibrinolysis which is hypothesized to occur almost immediately after injury [84].

In summary, while the demonstrated molecular interactions largely suggest that complement overactivation is associated with a state of increased thrombosis, it is unclear as to what the physiological effects of this could be in the setting of TBI.

In the following section, we refer briefly to the outline of an observational cohort study, currently in preparation, to explore the links between the two systems following TBI.

## 5. Observational Cohort Study: Outline

The CC-TBI study (Coagulation and complement activity in traumatic brain injury) is designed as a single-center population-based observational cohort study that will be conducted at the Karolinska University Hospital in Stockholm, Sweden, between 2020 and 2021. All adult patients with isolated severe TBI, who are admitted within three hours of injury, will be eligible for inclusion, excluding those pre-treated with anticoagulants or antiplatelet medication. Arterial blood samples will be obtained at 3, 5, 8, 12, 24, 48, and 72 hours after initial injury and then daily during the first two weeks of ICU stay. The blood samples will be analyzed for conventional coagulation parameters, fibrin degradation products, antithrombin, fibrinogen, soluble fibrin, platelet function (Multiplate), rotational thromboelastometry, the TAFI, prothrombin fragment 1+2, and complement components C3a, C5a, and the MAC. Temporal trajectories will be created for each blood test, and graphical trends between markers of hemostasis and complement activity will provide observational data points that could strengthen the evidence of crosstalk between the two systems. A more in-depth molecular analysis on the interaction between complement and platelet function will follow. This will be performed by evaluating the association between platelet function and complement-deposition on platelets through flowcytometry, as well as by complement-activating blood from TBI patients and measuring the change this invokes in platelet function. Lastly, complement activity will be correlated to lesion progression on serial imaging, as well as thromboembolic complications and functional outcome.

Our hypothesis is that increased complement activity will occur in patients with a prothrombotic phenotype following TBI, and those with decreased platelet aggregability will have a lower concentration of complement deposition on their platelet cell surfaces. Moreover, since there is often a recovery of platelet function after its initial decline [75], we believe that there will be a time-dependent variation in the amount of complement deposition on platelets that corresponds to the change in platelet function.

The Swedish Ethical Review Authority has approved the study (Dnr: 2019-01178). Design: A single-center prospective observational cohort study.Patients: Adults with severe isolated TBI who are admitted within 3 hours of injury. Methods: Blood samples taken at pre-specified time points during the first week of treatment. An analysis of complement activity and hemostasis will be performed. Main outcome: A correlation between temporal trajectories of complement activity and markers of hemostasis, as well as an association between complement deposition on platelets and platelet function.

## Figures and Tables

**Figure 1 ijms-21-01596-f001:**
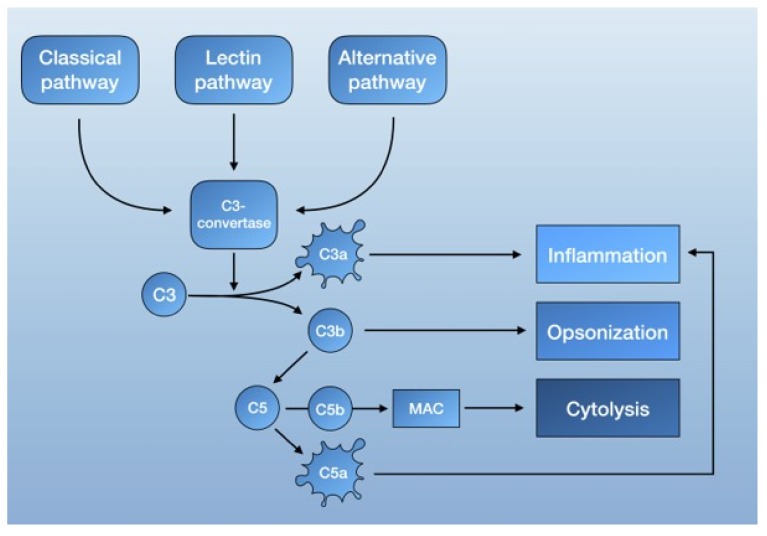
Schematic overview of the complement system.

**Figure 2 ijms-21-01596-f002:**
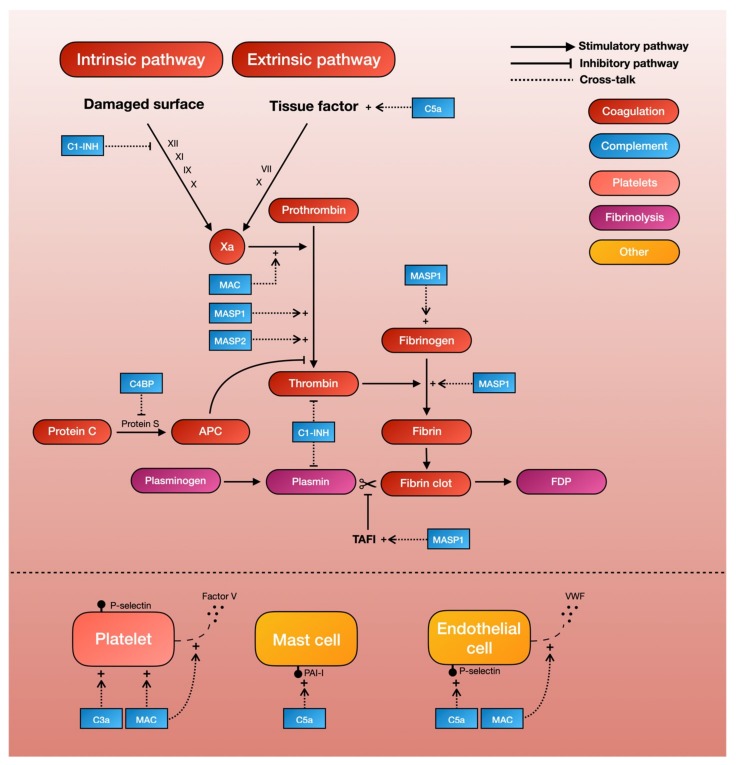
A schematic overview of complement-mediated coagulopathy.

**Table 1 ijms-21-01596-t001:** Complement-mediated coagulopathy: molecular interactions.

Complement Substrate	Effect on Hemostasis	Presumed Physiological Effect
C3a	Activate platelets	↑ Platelet aggregation
C5a	Increase tissue factor activity Increase expression of endothelial P-selectin Induce expression of PAI-1 on mast cells	↑Fibrin formation ↑ Platelet aggregation ↓ Fibrinolysis
MAC (C5b-9)	Activate platelets Initiates release of factor V from platelet alpha-granules Increase binding of coagulation factors Va and Xa Induce endothelial cells to secrete von Willebrand factor	↑ Platelet aggregation ↑ Platelet aggregation ↑Fibrin formation ↑Platelet aggregation, ↑ Fibrin formation
MASP 1	Activate thrombin (by cleaving prothrombin) Activate fibrinogen Activate factor XII Activate TAFI	↑ Fibrin formation ↑ Fibrin formation ↑ Fibrin formation ↓ Fibrinolysis
MASP 2	Activate thrombin (by cleaving prothrombin)	↑ Fibrin formation
C1-INH	Inhibit factor XII Inhibit thrombin Inhibit plasmin	↓ Fibrin formation ↓ Fibrin formation ↓ Fibrinolysis
C4BP	Inhibit protein S	↑ Fibrin formation

Abbreviations: C = complement factor; MAC = membrane attack complex; MASP = mannan-binding lectin serine protease; C1-INH = C1 esterase inhibitor; C4BP = C4b-binding protein; PAI-1 = plasminogen activator inhibitor-1; TAFI = thrombin-activatable fibrinolysis inhibitor.

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
