# Peer review of "Does Complement-Mediated Hemostatic Disturbance Occur in Traumatic Brain Injury? A Literature Review and Observational Study Protocol"

_ijms, 2020, doi:10.3390/ijms21051596_

Round 1

Reviewer 1 Report

In this review article, the authors provided evidence that following TBI, complement system is activated, which promotes thrombosis by interacting with coagulation cascade, platelets, fibrinolysis, and endothelia cells, contributing to poor functional outcomes. Additionally, they outlined a observational study to further evaluate the interaction between complement activity and coagulation in TBI patients.

Overall, except for some typos and grammatical errors, this review article is well written, related literatures are broadly and carefully reviewed, and information provided is very helpful for TBI treatment. I recommend it for publication.

Author Response

Thank you for reviewing this manuscript. Typos and grammatical errors have now been amended, and highlighted with “tracked changes”. Please let us know if you have any further queries.

Reviewer 2 Report

In this manuscript, the authors reviewed the mechanisms of complement-mediated hemostatic disturbance. This manuscript is very well summarized and interesting. However, there are a number of concerns as follow.

1) This manuscript focuses on the pathology of traumatic brain injury (TBI). Could you please explain what caused complement-mediated hemostatic disturbance in TBI patients. What triggers this reaction? This answer will make easier to image the clinical relevance.

2) The authors suggested complement-inhibiting drugs for treatment of this coagulation disorders in TBI patients. Complement is intricately intertwined with this coagulation disorder and has both positive and negative effects. It’s hard to imagine that complement-inhibiting drugs simply works. Could you please find out what works well and what is problems with complement-inhibiting drugs in TBI patients.

3) How about giving a specific TBI case in clinical examples of complement-mediated hemostatic disturbance?

Author Response

Thank you for reviewing this manuscript. We have addressed each comment point-by-point below. All changes made in the revised manuscript have been highlighted with “tracked changes”.

1) This manuscript focuses on the pathology of traumatic brain injury (TBI). Could you please explain what caused complement-mediated hemostatic disturbance in TBI patients. What triggers this reaction? This answer will make easier to image the clinical relevance.

Response: As highlighted in this review, there is increasing evidence supporting complement and coagulation amplification following TBI, as well as clinically significant interaction between the two systems in non-TBI cohorts. Potential links between the two systems in the given context are suggested and described. However, it is not known whether these interactions also exist in TBI. Therefore, we are not yet able to explain what might cause complement-mediated hemostatic disturbance in TBI patients. However, we do hope to add some clarification to this as part of the prospective observational cohort study outlined in this manuscript. This has now been clarified in the revised manuscript.

2) The authors suggested complement-inhibiting drugs for treatment of this coagulation disorders in TBI patients. Complement is intricately intertwined with this coagulation disorder and has both positive and negative effects. It’s hard to imagine that complement-inhibiting drugs simply works. Could you please find out what works well and what is problems with complement-inhibiting drugs in TBI patients.

Response: We have not suggested using complement-inhibiting drugs for the treatment of coagulation disorders in TBI patients. Instead, we speculate on the role that disproportionate complement activation may have in patients with TBI (lines 343 – 361). In this section we describe that, while the molecular interactions suggest that complement overactivation would be associated with a state of increased thrombosis, it is yet unclear as to what the physiological effects of this might be in the setting of TBI. Thus, until we know if complement-mediated coagulopathy exists and has clinical relevance in TBI, we believe it would be preemptive to speculate on a potential role for complement-inhibiting treatment. This has now been clarified in the revised manuscript.

3) How about giving a specific TBI case in clinical examples of complement-mediated hemostatic disturbance?

Response: We do not yet know if complement-mediated coagulopathy exists and has clinical relevance in TBI, and are therefore not able to describe a specific TBI-related case. Clinical examples of complement-mediated hemostatic disturbances have been described in the context of other non-TBI entities and are presented in section 3.5. of the manuscript. We do hope to be able to provide more insights and potentially specific TBI cases upon completion of the observational cohort study described in the manuscript. This has now been clarified in the revised manuscript.

Please let us know if you have any further queries.

This manuscript is a resubmission of an earlier submission. The following is a list of the peer review reports and author responses from that submission.